# Signals of complexity and fragmentation in accelerometer data

Els Weinans[1]*, Jerrald L. Rector[2]*, Sarah Charman[3], Renae J. Stefanetti[4,5], Cecilia Jimenez-Moreno[3], Gráinne S. Gorman[4,5], Ingrid van de Leemput[6], Daniël van As[7], René Melis[2], Baziel van Engelen[7]

1 Copernicus Institute of Sustainable Development, Environmental Science, Faculty of Geosciences, Utrecht University, Utrecht, The Netherlands, 2 Department of Geriatrics, Radboud Institute for Health Sciences, Radboud University Medical Centre, Nijmegen, The Netherlands, 3 Cardiovascular Research Centre, Institutes of Cellular and Genetic Medicine, Newcastle University, Newcastle upon Tyne, United Kingdom, 4 Welcome Centre for Mitochondrial Research, Faculty of Medical Sciences, Translational and Clinical Research Institute, Newcastle University, Newcastle upon Tyne, United Kingdom, 5 National Institute for Health and Care Research (NIHR) Newcastle Biomedical Research Centre (BRC), Newcastle upon Tyne, United Kingdom, 6 Aquatic Ecology and Water Quality Management, Wageningen University and Research, Wageningen, The Netherlands, 7 Department of Neurology, Donders Institute for Brain Cognition and Behaviour, Radboud University Medical Centre, Nijmegen, The Netherlands

☯ These authors contributed equally to this work.
* e.weinans@uu.nl (EW); jerraldrector@hotmail.com (JLR)

**Data availability statement:** The raw accelerometer data and the code to reproduce

## Abstract

There is a growing interest to analyze physiological data from a complex systems perspective. Accelerometer data is one type of data that is easy to obtain but often difficult to analyze for insights beyond basic levels of description. Previous work hypothesizes that an individual's activity pattern can be seen as a complex dynamical system. Here, we explore this hypothesis further by investigating whether complexity-based measures quantifying repetitiveness and fragmentation of activity captured via accelerometer can detect health differences beyond traditional measures. Our results demonstrate that healthy individuals have a higher regularity (indicated by a lower correlation dimension), a higher probability of activity after a period of rest, and a lower probability of a period of rest after a period of activity compared with patients living with Myotonic Dystrophy type I (DM1), a chronic, progressive, complex, multisystem disease. For the correlation dimension, this difference was independent of the average, coefficient of variation and autocorrelation of the activity signals. This suggests that the correlation dimension can extract clinically relevant information from accelerometer data. Therefore, our results corroborate the idea that a complexity perspective may help to reveal the emergent characteristics of health and disease.

## Introduction

Conceptualizing and quantifying health is crucial to assess health policies [1]. Under conditions of health, an individual is normally capable of conducting activities of daily living without undue constraints imposed by their environment or internal limitations arising from disease processes. Indeed, health is often conceptualized as freedom to choose how to live and

the figures is available from https://github.com/elsweinans/optimistic_data_code.

**Funding:** Renae J. Stefanetti is supported by the National Institute for Health and Care Research (NIHR) Newcastle Biomedical Research Centre (BRC). The NIHR Newcastle BRC is a partnership between Newcastle Hospitals NHS Foundation Trust, Newcastle University and Cumbria, Northumberland, Tyne and Wear NHS Foundation Trust funded by the National Institute for Health and Care Research (NIHR). The views expressed are those of the author(s) and not necessarily those of the NIHR or the Department of Health and Social Care.

**Competing interests:** The authors have declared that no competing interests exist.

take control of one's own actions [2]. This control can only be gained by complex feedback mechanisms, e.g. a pile of rocks will follow the temperature of the environment whereas a mammal controls their own body temperature. Therefore, there is growing interest in studying health care problems from a 'complex systems perspective' [3–7]. In this approach, instead of studying individual diseases, emphasis is given to the underlying interactions of physiological, behavioral, or psychological processes and their dynamical properties. In this study, we explore whether two proposed complexity-based measures can capture meaningful dynamical information: the correlation dimension [8] and fragmentation indices [9].

Of the many physiological time series commonly captured by wearable sensors, physical activity measured by accelerometry has garnered substantial attention by researchers looking to gain insight into the health functioning of the individual [10]. Indeed, objectively measured physical activity is a well-known indicator of human health [11] that has been shown to more reliably capture activity patterns than self-reported questionnaires [12–14]. The dynamics of accelerometer time series data is unique in the fact that it integrates information from a broad set of physical, mental and social interactions. The integrated factors range from mood and well-being [15] to cardiovascular functioning and environmental conditions, as well as mobility, balance, and muscle strength [16,17]. This holistic characteristic makes it particularly attractive to analyze the multi-faceted components of health by exploring dynamical signal properties.

The field of complexity science has provided numerous measures that capture such dynamical properties of the signal of interest. There are two classes of measures from complexity science that we believe have particular relevance for capturing the complex processes that characterize healthy activity dynamics. The first includes measures that look at patterns of repeatability, such the correlation dimension [8], approximate entropy [18], sample entropy [19], multiscale sample entropy [20], and recurrence plots [21]. The second includes fragmentation analysis alternatives to transition probabilities, such as the alpha and GINI index of sedentary time accumulation [22] or time spent in prolonged sedentary bouts [23]. Here, we select the correlation dimension from the first class and activity fragmentation based on state transition probabilities from the second. By our decision, we are not endorsing any particular method, but aim to illustrate the utility of 'complexity methods' in general for analyzing accelerometer data.

The correlation dimension is linked to the 'fractal dimension' of the dynamical system, i.e. it describes the patterns and shape of the signal. It does so, by calculating patterns of repeated sequences in the data. If the time series are highly repetitive, the correlation dimension is low, if the time series are completely random, the correlation dimension is high [8] (see Methods section for a more extensive explanation). The correlation dimension, and its various extensions, have been applied to numerous physiological datasets, including EEG signals (changes during absence epilepsy [24], distinguishing Alzheimer patients from healthy controls [25], and distinguishing different sleep stages [26]), electrocorticography signals (identifying distinct brain states in monkeys [27]) heart rate variability (distinguishing between dilated cardiomyopathy patients and healthy controls [28]) and muscle activity (classifying number of balls used by a juggler [29]).

Fragmentation analysis reveals how fragmented or consolidated an individual's activity-rest patterns are. These measures were designed to explicitly consider both the duration of periods of physical activity and their distribution over the course of the day. Previous work demonstrates that not only total sedentary time, but also the pattern in which a person accumulates sedentary time, is indicative of a person's health [30]. These patterns are calculated based on a probabilistic state transition model. Briefly, this approach models the observed

data as a series of transitions between discrete states and describes the frequency and distribution of such transitions. In health-related applications, it has been used to discern the temporal organization of behavioral states from polysomnographic data, and more recently, actigraphy data [9,31]. In the case of physical activity, $k_{AR}$ represents the probability of an individual to transition to a resting state after sustaining activity for a given period, while $k_{RA}$ is associated with the probability that they transition from a sustained resting state to an active state.

Here, we investigate whether correlation dimension analysis and fragmentation analysis can extract clinically relevant information from accelerometer data. In line with Glass & Kaplan (1993) [32], we evaluate the 'clinical relevance' of the measures based on two criteria:

1. They should be able to distinguish between groups with known differences in health, and
2. They should provide unique information, i.e., information that is different from traditional or simpler analysis tools.

To test the first criterion, we performed correlation dimension analysis and fragmentation analysis on accelerometer data of healthy individuals and patients with DM1. DM1 is a chronic, progresive, genetic condition characterized by premature aging and loss of resilience [33]. It is the most common adult muscular dystrophy and is regarded as a multisystemic disorder as it involves brain, muscle, heart, gut, eyes, endocrine and immune systems [34] that leads to "severe physical impairment, restricted social participation, and premature death"[33]. This patient population thus represents individuals with known differences in health status against whom to explore measures that capture the impact of activity constraints imposed by disease. To test the second criterion, we compared the output of the correlation dimension and fragmentation analysis to the average, the coefficient of variation, and the lag-1 autocorrelation of the activity. These measures are common tools to analyze time series and can give general insights into an individual's well-being [4,32,35]. As these 'traditional' measures are a lot easier to apply than the correlation dimension or fragmentation analysis, we require our complexity measures to extract information above and beyond the information that is already obtained by the traditional analysis methods.

## Materials and methods

### Data and pre-processing

For this study, we re-analyzed accelerometer data from the OPTIMISTIC trial [33,36] (ClinicalTrials.gov number: NCT02118779). For this trial, data from DM1 patients were obtained at four neuromuscular referral centres in Paris (France), Munich (Germany), Nijmegen (The Netherlands) and Newcastle (UK). To avoid confounding by the intervention of the OPTIMISTIC trial, we only used baseline data. This lead to a group of 59 individuals with a mean age of 46.54 years (std 11.95 years) of which 54% were male. The accelerometer data was obtained via a triaxial accelerometer (GENEActiv, ActivInsights Ltd, UK) which was ankle worn and gave a signal every 5 seconds. The data was available for 14 consecutive days, including day and night.

We compared accelerometer data from the DM1 group to accelerometer data from a group of healthy controls recruited from Newcastle University, Newcastle upon Tyne, United Kingdom, using the same type of device [37]. Newcastle University Ethics committee approved the study (Ref: 5572/2016). This group consisted of 19 adults with a mean age of 35.32 years (std 10.32 years) of which 42% were male. The data was available for 7 consecutive days. In

our main analysis we ran all analysis on the full length datasets. In our sensitivity tests, we tested whether the uneven lengths between the DM1 group and the healthy group affected our results.

All patients and healthy controls provided written informed consent. All procedures conducted in this study were in accordance with the Declaration of Helsinki. The data was accessed between October and December 2020. Authors did not have access to information that could identify individual participants.

In our main analysis, we course grained the data to 1 minute bouts by averaging over blocks of 12 data points to capture dynamics on the minute scale. We later also tested the sensitivity of this choice by repeating the analysis for different time scales.

## Traditional analysis

The most used and most intuitive measure to obtain from any signal is the average value. For accelerometer data, the average activity indeed manages to detect differences between healthy people, people with chronic fatigue syndrome and people with multiple sclerosis, where healthy subjects are characterized by a high average activity [38]. Here, we calculate the average activity over the preprocessed time series. In addition, we calculate the standard deviation and the coefficient of variation (standard deviation divided by the mean) of the time series. Last, in line with recommendations [32,39], we calculate the autocorrelation at lag-1 (which means the autocorrelation at a lag of 1 minute) as an additional measures that could capture interesting properties of the signal.

## Correlation dimension analysis

To obtain a measure of the complexity of the signal (i.e. the irregularity of a signal), we calculate the correlation dimension. The correlation dimension captures signals of repetition in the data. Generally speaking, highly repetitive signals yield a low correlation dimension and highly irregular signals yield a high correlation dimension. The correlation dimension is a measure of the dimensionality of the space that is covered by a dynamical system and is as such often seen as a 'fractal dimension' of a signal [39]. However, here we use it more loosely as a complexity measure that captures patterns of irregularity in a time series.

To calculate the correlation dimension, first the correlation sum or correlation integral should be calculated. The correlation integral $C_d$ is defined as the mean probability that segments of a time series repeat themselves. If we assume a time series $[x_1, x_2, x_3, ..., x_n]$, then $C_d$ can be calculated as

$$C_d(r) = \lim_{N \to \infty} \frac{2}{N(N-1)} \sum_{i=1}^{N} \sum_{j>i}^{N} \theta(r - |X_i - X_j|),  \tag{1}$$

Where the $\theta$ is the heaviside function and $X_i$ is a vector of consecutive values of the time series $x_i, x_{i+1}, ..., x_{i+d}$ where $d$ is the 'embedding dimension'. The parameter $r$ determines what constitutes points that are 'close together', i.e. when it is said that a segment of the time series is repeated. The parameter $N$ is the number of points in $X$ (more precisely, $N = n - d + 1$). Multiplying by $\frac{2}{N(N-1)}$ ensures that $C_d(r)$ calculates the average fraction of neighbors that each point has for a particular $r$, excluding self-matches [39]. $|X_i - X_j|$ is a distance between the vectors $X_i$ and $X_j$, here by default we use the euclidean distance, in line with [8]. $C_d(r)$ follows a power law for small r where

$$C_d(r) \sim r^\nu,  \tag{2}$$

where $\nu$ is the correlation dimension of the time series [8]. The value of $\nu$ can be obtained by plotting $\log(C_d(r))$ against $\log(r)$ and calculating the slope of a linear fit (Fig 2).

Graphical representations of the steps are visualized in Figs 1 and 2.

There are two important considerations to keep in mind when using the correlation dimension. Firstly, the link to the fractal dimension is clear for deterministic, chaotic attractors, but this link is not straightforward to the noisy time series we are investigating here [39]. Secondly, calculating correlation dimension requires a number of free parameters to be chosen. These two considerations are linked: For deterministic, chaotic, low-dimensional attractors such as the Hénon map or the lorenz attractor, as soon as the 'true dimension' of the attractor is reached, the results from the correlation dimension analysis are independent of $r$ and $d$. In these cases, the correlation dimension analysis thus provides a property of the time series that had a strong theoretical foundation and meaning. In contrast, for non-deterministic systems that might not be (purely) governed by a low-dimensional chaotic attractor, the values obtained from the correlation dimension analysis are not inherent, stable properties of the signal, but depend on the choice of the parameters [32,39] (see S1 Text Sect 4 for additional explanation and figures). Therefore, in this case, the measure captures relative instead of absolute properties of the signal [40,41]. Different signals can be compared to each other only when they are calculated with the same parameter values [41]. If we denote the correlation dimension of time series X where $r = r_1$ and $d = d_1$ as $\nu(X)_{d1}^{r1}$ and we find that $\nu(X)_{d1}^{r1} < \nu(Y)_{d1}^{r1}$, then $\nu(X)_{d2}^{r2} < \nu(Y)_{d2}^{r2}$ should hold as well (i.e. even though absolute values do change for different parameter values, qualitatively the results should hold) [19,40].

## Parameter selection

We set our parameters in line with recommendations from the literature [40,42]. Since the choice of parameters in the correlation dimension analysis is a subject of debate [43,44], we performed extensive sensitivity tests to explore the sensitivity of our results to deviations in our default parameters.

To choose default settings for the parameters $r$ and $d$, we followed recommendations from previous studies. In work dealing with the sample entropy (a measure that starts with the $C_d(r, d)$ plot of Fig 2, and then looks at the distance between the $C_d(r, d)$ and $C_d(r, d + 1)$ for a fixed $r$) the parameter $r$, measured in standard deviations of the data, is often set to 0.2 as a default [40]. Here, we selected a range of 0.03 - 0.3 to ensure that the default of 0.2 was within the range of our analysis (see S1 Text, Sect 2.1 for additional explanation and visualization). The parameter $d$ was set at 6 as for most healthy individuals the increase in the correlation dimension for increasing values of $d$ seemed to have converged around 6. For DM1 individuals, no convergence was found even for higher values of $d$, similar to what can be observed in random noise data (see S4 and S5 Figs).

To evaluate the robustness of our results, in line with recommendations [42], we tested the sensitivity to the chosen parameters by rerunning our analysis for different parameter settings (see S1 Text, Sect 5). Our free parameters are the distance within which we look for neighboring points ($r$), the embedding dimension ($d$), and the time interval between the data points used in $X$ (we call this time interval $\tau$). Furthermore, in the preprocessing step of the data, we could choose the time scale in which we wanted to analyse our data, i.e., whether we wanted to look at changes between 5 seconds, between minutes or between hours. We also tested the sensitivity to this choice. Lastly, in our analysis we used the Euclidean distance to calculate

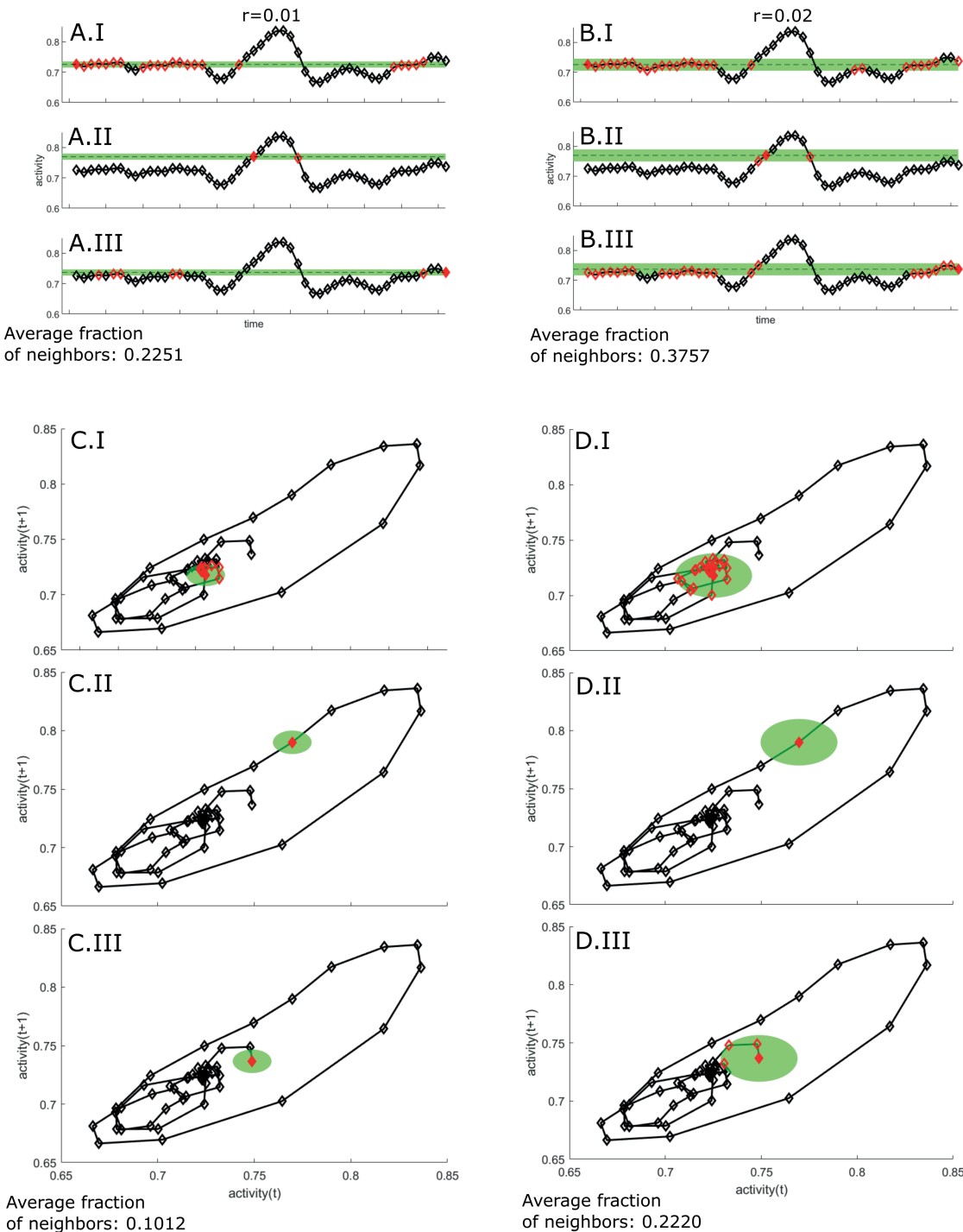

Average fraction
of neighbors: 0.2251

Average fraction
of neighbors: 0.3757

Average fraction
of neighbors: 0.1012

Average fraction
of neighbors: 0.2220

**Fig 1. Calculating the correlation integral from a time series.** A and B illustrate the time series with a diamond for every data point for $d = 1$ (dimension 1). For a fixed $r$, we denote for every point how many points are within a distance $r$. The green area depicts the distance of $r$ from the filled diamonds (points 1 (panels I), 25 (panels II) and 51 (panels III)). All red diamonds are within the distance r from the filled diamond (i.e. they are within the green shaded area). The average fraction of points that are neighbors are 0.2251 for $r = 0.01$ and 0.3757 for $r = 0.02$. Next, in C and D, we calculate the correlation integral for $d = 2$ (dimension 2). Here, we do not just look if a point is visited repeatedly, but we look if sets of 2 consecutive points are visited repeatedly. We plot every point in the time series against the next point of the time series (C and D). The green shaded ares shows the area for which points are considered neighbors for point 1 (panels I), point 25 (panels II) and point 51 (panels III). As $r$ increases from 0.01 to 0.02 (from C to D), the average fraction of neighbors increases from 0.1012 to 0.2220.

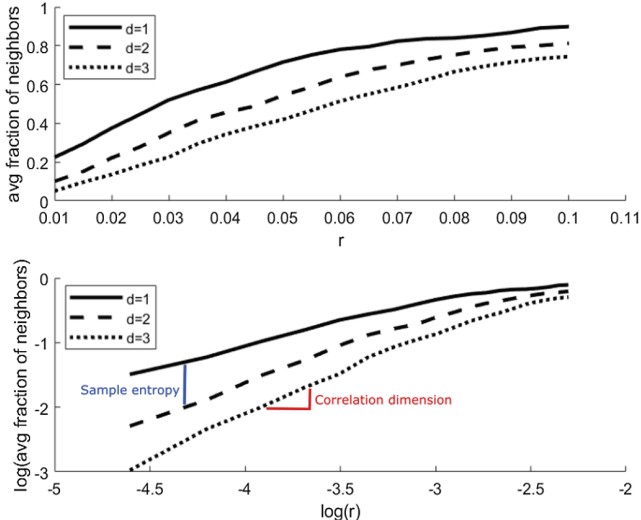

**Fig 2. The correlation integral for various values of *r* and *d* form the foundation for other complexity measures such as correlation dimension and sample entropy.** Top panel illustrates the development of the fraction of neighbors as *r* and *d* increase. Bottom panel shows the same but on log-log scale. The correlation dimension is defined as the slope of the line as r increases. The sample entropy is defined as the distance between two lines as d increases. The points from Fig 1 are all on these lines.

$|X_i – X_j|$, in line with previous work [8]. Later extensions to the correlation dimension suggested to use the Chebyshev distance (i.e. the maximum distance) instead, mostly for computational reasons [19]. To test if our choice affected our results we repeated our analysis using the Chebyshev distance.

### Fragmentation analysis

The fragmentation of rest and activity was quantified using state transition measures developed and validated by Lim and colleagues: the active-rest transition probability ($k_{AR}$) and the rest-active transition probability ($k_{RA}$) [9]. Respectively, these measures reflect the probability that an individual will rest after sustaining activity for a certain period of time and the probability of becoming active after sustaining rest.

First, the pre-processed accelerometer data was coded as one of the binary states of rest or activity, based on whether its value was above or below a cut-off value of 0.0167 *g*, chosen based on visual inspection (see Fig 3A). While $k_{AR}$ and $k_{RA}$ were found to be relatively insensitive to the selected activity threshold [9], this cut-off value exceeded the baseline noise level of our raw accelerometer data. Runs of rest were defined as periods beginning with at least one rest epoch and ending at the epoch before the first epoch of activity. Runs of activity were similarly defined as beginning with at least one active epoch and ending before the first rest epoch. Thus, each rest or active run begins and ends with an active-rest or rest-active transition. Then, the conditional probability that an individual would be resting at time $t$+1 given that they were active for the preceding $t$ epochs was defined as $pAR(t) = P(R|At)$. $pRA(t) = P(A|Rt)$ was similarly defined as the conditional probability of becoming active at time $t$+1 given that they were resting for the previous $t$ epochs. $N_t$ was defined as the number of runs, $N$, of duration $\geq t$ for $t = 1, 2,... t_{max}$. The proportion was calculated by dividing the resulting area under the curve by the total number of observed active or rest bouts.

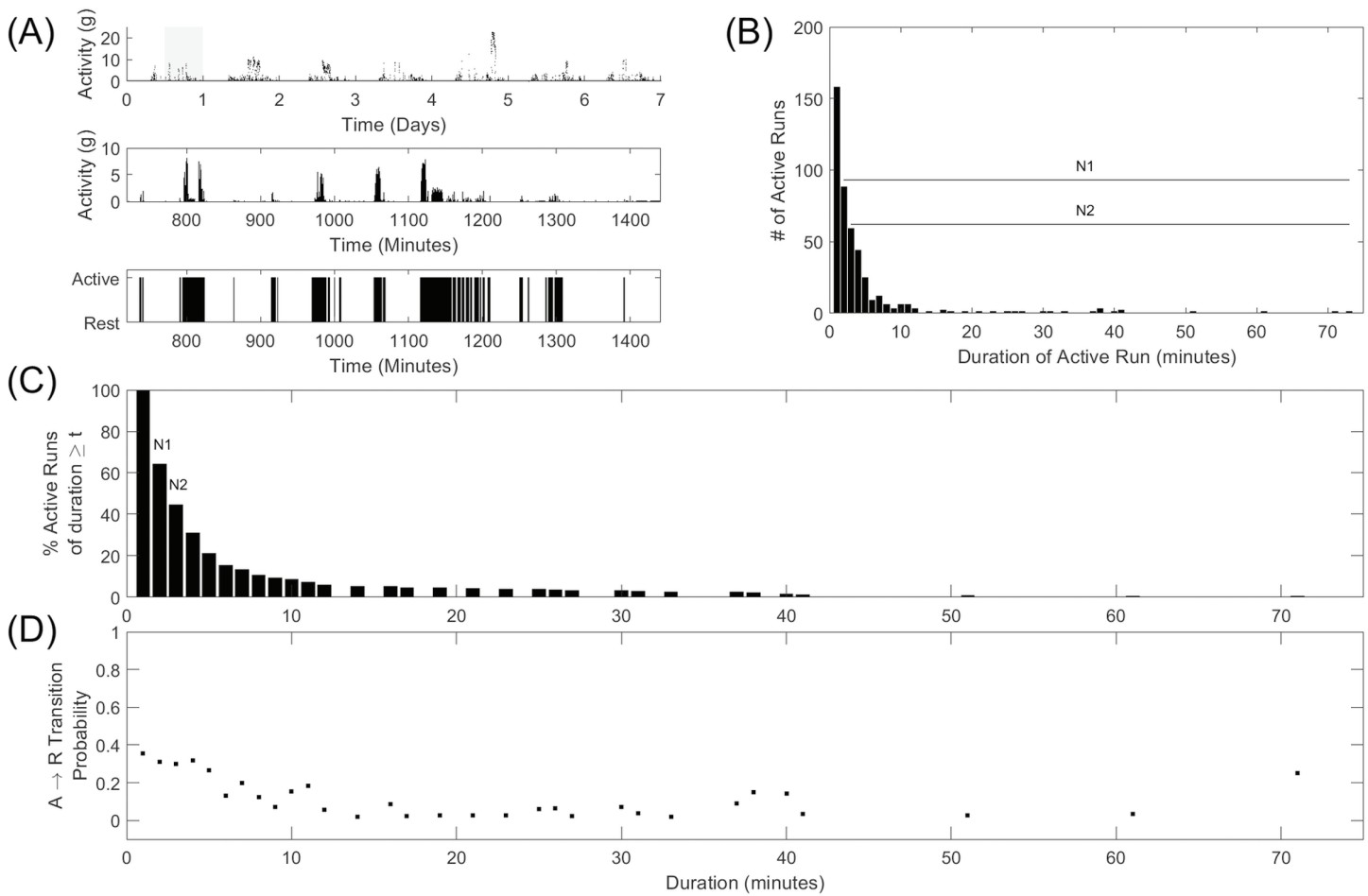

**Fig 3. Calculation of transition probability $pAR(t)$ for a single healthy participant.** (A) Shows 7 days of accelerometer data; a zoom-in of a 12-hour period (shaded region above); and the binary active-rest classification. (B) Shows a histogram of active bout durations observed across the 7 days. (C) Shows the number of active runs, $N$, of duration $\geq t$ for t = 1, 2,..., $t_{max}$. The values of $N_1$ and $N_2$ as plotted in panel C are the areas under the curve in panel B from times 1 and 2 onward, respectively as indicated by the black lines in panel B. (D) Shows the active-rest transition probabilities, $pAR(t)$, after being active for each duration. Adapted from [9].

As not all run lengths, $t$, may be represented for an individual, the equation for $pRA(t)$ was modified to

$$pRA(t) = \frac{N_t - N_{t+d}}{N_t d}, \tag{3}$$

where $d$ is the time interval until the next represented $t$. Here, high transition probabilities reflect greater fragmentation of rest and activity characterized by short runs of rest and activity, whereas low transition probabilities reflect more consolidation and longer runs of rest and activity. The same was done for $pAR(t)$.

The temporal trends in $pRA(t)$ and $pAR(t)$ followed a similar pattern as originally described whereby three regions could be distinguished [9]. The constant region was operationally defined as the longest stretch within which the LOWESS curve fit to the $pRA(t)$ and $pAR(t)$ data varied by no more than 1 standard deviation of the corresponding $pRA(t)$ or $pAR(t)$ curve. The falling and rising regions were defined as the regions before and after the constant region, respectively. $k_{RA}$ and $k_{AR}$ were defined as a weighted mean of the $pRA(t)$ and $pAR(t)$ transition probability estimates within the constant region for each curve. Weights

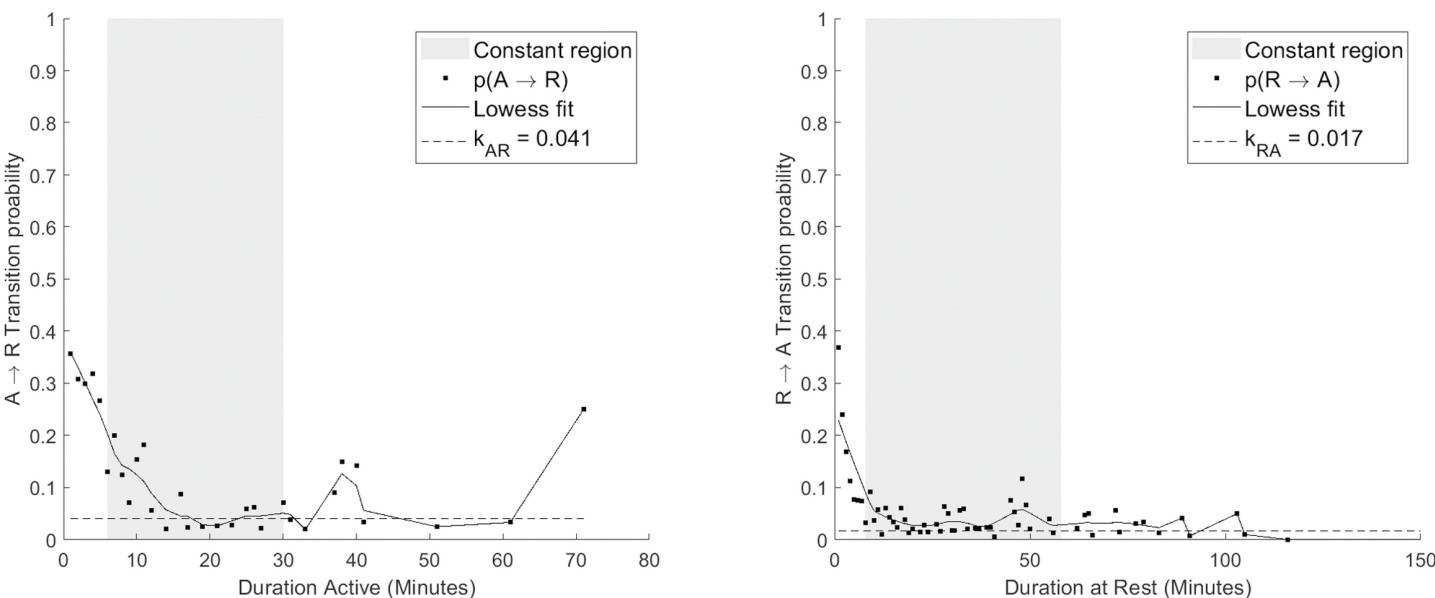

**Fig 4. Extracting fragmentation values.** Using the transition probabilities resulting from the procedure in Fig 3, the rest-active ($k_{RA}$) and active-rest ($k_{AR}$) fragmentation values are calculated by taking the average of the transition probabilities within the constant region – defined as the longest stretch within which the LOWESS curve fit to the $pAR(t)$ or $pRA(t)$ data varied by no more than 1 $SD$ of their respective curves – weighted by the number of observations used to estimate the probability. Adapted from [9].

were proportional to the square root of the total number of runs contributing to each probability estimate [9].

Together, $k_{RA}$ and $k_{AR}$ can give insight into the strategy and pattern of sedentary time accumulation. These patterns are not explicitly captured in traditional analyses that look at means, variances, or time spent in various intensities of activity. However, it is expected to overlap with temporal auto-correlation, which captures how likely an increase or decrease in activity is to be followed by the same. The fragmentation of sedentary behavior is preferable over activity *perse* as focusing only on active periods necessarily excludes sleeping periods (which contribute heavily to total sedentary time) and must use a priori assumptions about waking hours [22].

## Statistical analysis

First bi-variate correlations were used to assess the potential overlap in information provided by the traditional and dynamic measures.

Next, group comparisons of the measure between patients with DM1 and healthy controls were carried out using Mann-Whitney U tests.

Finally, traditional and dynamic measures were compared between patients and healthy controls using binary logistic regression analysis. A binary measure of group membership (i.e., healthy = 1; DM1 = 0) was used as the dependent variable. First, a model was created as a baseline logistic regression model that corrects for age, sex, average activity, coefficient of variation and lag-1 autocorrelation of activity (Model 1). Next, each of the complexity measures (correlation dimension, $k_{AR}$, and $k_{RA}$) were entered in separate models as independent

predictors of group membership (Models 2-4). Values for $k_{AR}$, and $k_{RA}$ were first logit transformed to approximate normality and converted to z-scores. Extreme values (> ± 2 SD) were removed before analysis.

To assess whether the inclusion of the complexity measure improves the model enough to warrant the increase in explanatory variables, we determine the Akaike and Bayesian information criteria (AIC and BIC respectively). The inclusion of a variable is warranted if the information criterion of the larger model is lower than the information criterion of the smaller model [45].

## Results

Correlations between the traditional and dynamic indicators across all participants were generally moderate, but highly significant, with absolute correlations ranging from $\rho$ = 0.2 to 0.7 (Table 2, S1 Fig). One exception was the correlation between the mean and standard deviation at $\rho$ = 0.93. Accordingly, the coefficient of variation (CoV) was used in subsequent analyses.

Compared to patients with DM1, healthy individuals displayed a higher average activity, higher variation in their dynamics, and higher autocorrelation (Mann-Whitney U = 1910, U = 2037, and U = 1849, respectively; all p<.001, Fig 5). Further, healthy individuals showed a lower correlation dimension (U = 2857, p<.001), a lower $k_{AR}$ (U = 2688, p<.001), and a higher $k_{RA}$ (U = 2101, p<.001; Fig 6).

Binary logistic regression models revealed that the correlation dimension adds significantly to the model that adjusts for age, sex, average activity, coefficient of variation, and autocorrelation (see Table 2). Every 1 SD increase in correlation dimension was associated with 99% reduced odds of being in the healthy group (OR [95% CI] = 0.0069 [0.00018 - 0.268], p=0.00076; Model 2). The state transition probabilities are not statistically significant, with the effect of $k_{AR}$ at a 67% odds reduction (OR = 0.33 [0.11 - 0.99], p=.065; Model 3), and the effect of $k_{RA}$ at a two-fold increase in odds (OR = 2.2 [0.7 - 7.11], p=0.17; Model 4). The Akaike

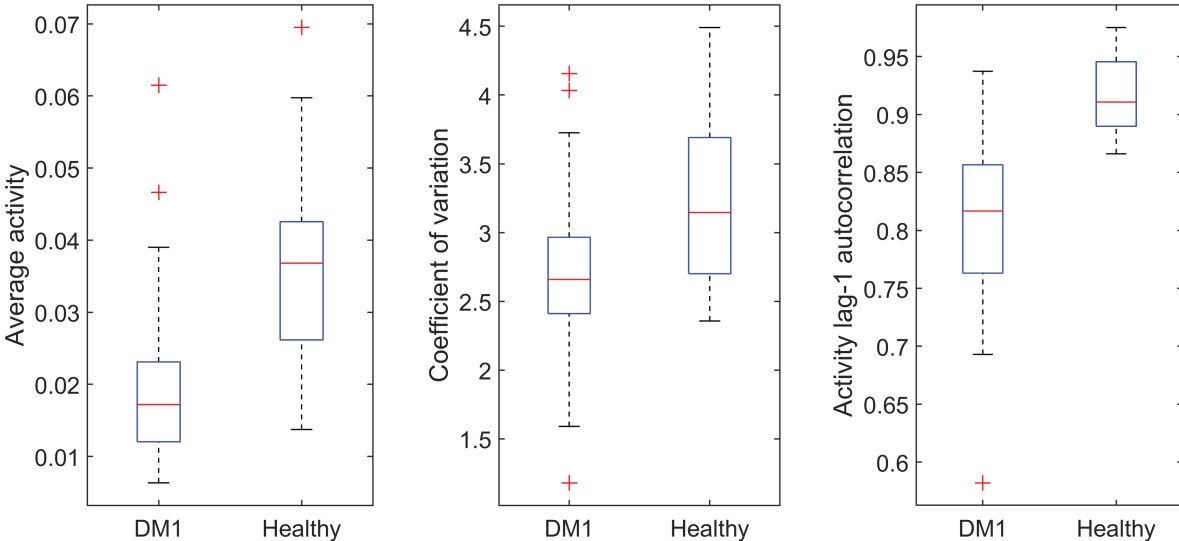

**Fig 5. Traditional analysis of accelerometer data: Average, coefficient of variation and autocorrelation** Healthy individuals have a higher average activity, a higher coefficient of variation, and a higher lag-1 autocorrelation than individuals with Myotonic Dystrophy type I.

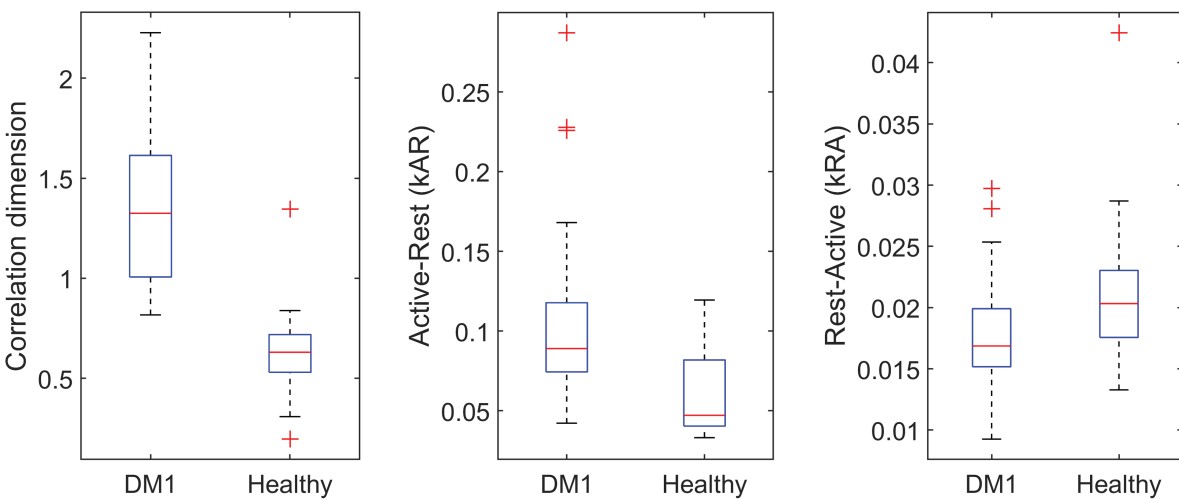

**Fig 6. Dynamic indicators for healthy individuals and individuals with Myotonic Dystrophy type I.** Healthy individuals have a lower correlation dimension, a lower active-to-rest transition probability and a higher rest-to-active transition probability than individuals with Myotonic Dystrophy type I.

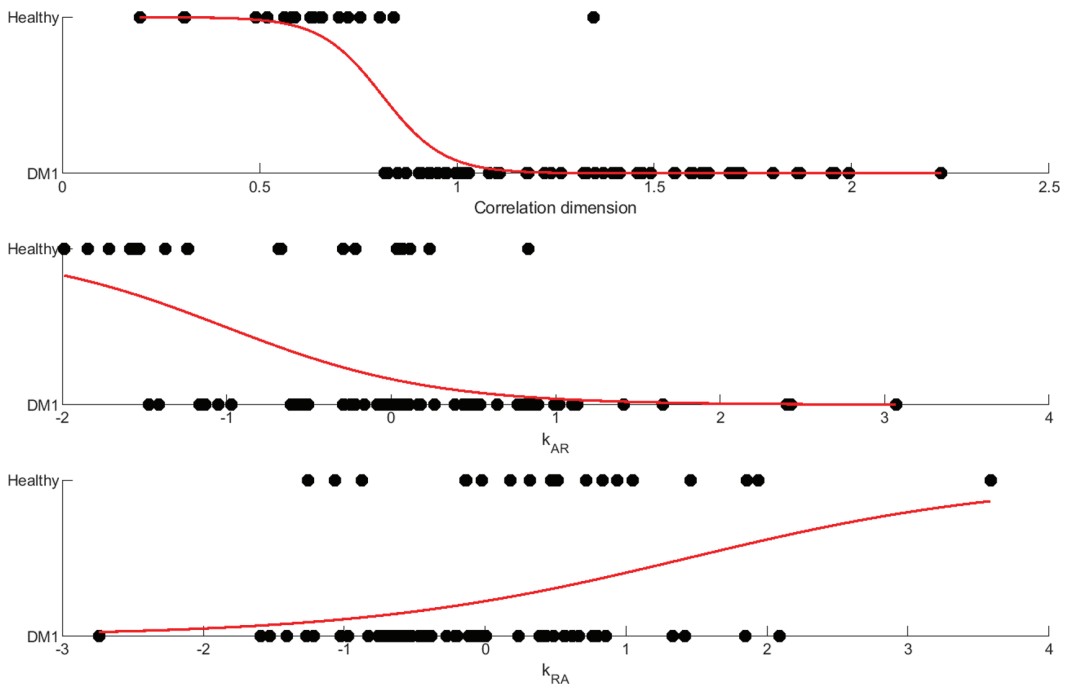

**Fig 7. Binary logistic regression for correlation dimension, $k_{AR}$ and $k_{RA}$,** the factors that constitute the logistic model that distinguishes between healthy subjects and patients with myotonic dystrophy type I.

information criterion suggests that all complexity measures improved the performance of the basic model without complexity measures. The Bayesian information criterion suggests that model 2 (with the inclusion of the correlation dimension) is an improvement on the basic model, model 3 (with the inclusion of kAR) is equally likely as the basic model, and model 4 is less likely than the basic model (Table 2).

Since our two groups differ in age and sex, and there are possible collinearity effects present (see Table 1), we repeated the analysis based on a subsample of our DM1 group that was created to match the healthy group with respect to age and sex. In this subsample, the correlation dimension and kAR differ significantly between the healthy group and the DM1 group, but, kRA is no longer significant (S3 Fig). We also repeated the logistic regression analysis for the reduced sample, but this re-analysis suffered from the reduced amount of data (and thus reduced statistical power) and therefore we cannot draw conclusions from this analysis. Nevertheless, all effects are in the same direction as in our main analysis and AIC and BIC support the inclusion of the correlation dimension and kAR in the logistic regression models (see S1 Table).

To test whether our parameter selection influenced our results, we repeated our analysis with different parameter settings. Changes in parameter selection did not alter our conclusions (see S1 Text: Sects 5 and 7). Absolute values for the correlation dimension do change for different values of the embedding dimension, range of $r$, time scale, time series length and distance metric that is used (see S7, S8, S10, S11, S12, S13 Figs), but qualitatively the results lead to the same conclusions. Therefore, when different studies are compared to each other, these dependencies should be taken into account. Sensitivity tests for the fragmentation measure varying the chosen epoch length showed qualitative similar patterns between the groups and values of $k_{AR}$ and $k_{RA}$. The calculated values of $k_{AR}$ using various epoch length were highly significantly correlated, and with the exception of the 5 second-30 minute correlation (p = .181), those for $k_{RA}$ also significantly correlated with one another (see S2 Table and S13 Fig). These results suggest relative insensitivity of the fragmentation measures to the choice of epoch length at higher time resolutions, confirming previous results [9].

**Table 1. Pearson correlations between indicators, with the p-value indicated between brackets. See S1 Fig for a visualization.**

|            | avgs          | stds          | CoV           | AC            | corrdims      | $k_{AR}$        |
|------------|---------------|---------------|---------------|---------------|---------------|---------------|
| stds       | 0.938 (0)     | -             | -             | -             | -             | -             |
| CoV        | 0.211 (0.063) | 0.502 (0)     | -             | -             | -             | -             |
| AC         | 0.631 (0)     | 0.77 (0)      | 0.678 (0)     | -             | -             | -             |
| corrdim    | -0.544 (0)    | -0.652 (0)    | -0.468 (0)    | -0.633 (0)    | -             | -             |
| $k_{AR}$     | -0.584 (0)    | -0.629 (0)    | -0.39 (0)     | -0.597 (0)    | 0.489 (0)     | -             |
| $k_{RA}$     | 0.345 (0.002) | 0.303 (0.007) | 0.049 (0.671) | 0.252 (0.026) | -0.342 (0.002)| -0.306 (0.006)|

**Table 2. Binary logistic regressions for the full set of traditional analysis (model 1) and the extension of this model with either the correlation dimension (model 2), kAR (model 3), and kRA (model 4). Low p-values are indicated with $^o$ (p < 0.1), $^*$ (p < 0.05), or $^{**}$ (p < 0.001). The Akaike Information Criterion and Bayesian Information Criterion are calculated in rows AIC and BIC.**

|       | Model 1            | Model 2                        | Model 3            | Model 4              |
|-------|--------------------|--------------------------------|--------------------|----------------------|
| Age   | 0.96 (0.9 - 1.03)  | 0.95 (0.86 - 1.06)             | 0.95 (0.88 - 1.02) | 0.96 (0.017 - 0.528) |
| Sex   | 0.31 (0.055 - 1.74)| 1.5 (0.12-18.4)                | 0.31 (0.049 - 1.89)| 0.3 (0.049 - 1.78)   |
| Avg   | 2.2 (0.64 - 7.28)  | 2.2 (0.29 - 16.6)              | 1.9 (0.44 - 8.48)  | 2.3 (0.66 - 8.27)    |
| CoV   | 1.3 (0.21 - 8.73)  | 1 (0.028 - 36.1)               | 1.7 (0.18 - 16.6)  | 3.1 (0.32 - 30.7)    |
| AC    | 13 (1.4 - 125)$^*$   | 2.8 (0.16 - 51)                | 14 (1.1 - 178)$^*$   | 7.7 (0.83 - 72.1)$^o$  |
| Cd    | -                  | 0.0069 (0.00018 - 0.268)$^{**}$  | -                  | -                    |
| kAR   | -                  | -                              | 0.33 (0.1 - 1.07)$^o$| -                    |
| kRA   | -                  | -                              | -                  | 2.2 (0.7 - 7.11)     |
| AIC   | 50.1580            | 35.4125                        | 48.0306            | 49.3373              |
| BIC   | 64.2982            | 51.9095                        | 64.2530            | 65.5597              |

## Discussion

Our results show that correlation dimension analysis and fragmentation analysis can reveal information about the accelerometer signal that goes above and beyond the traditional statistical tools. Thus, together, they give a broader view of the dynamical properties of the signal. More specifically, they identify how an individual's activity is repeating itself (correlation dimension) and how the activity- and rest bouts are distributed (fragmentation analysis). We evaluated the clinical relevance of these measures based on two criteria: 1) can the measures distinguish between two groups with known differences in health? and 2) is the information provided from these measures unique compared to simpler measures? Compared to individuals with DM1, we found that healthy individuals had significantly lower correlation dimensions (indicating a more predictive pattern, as described by [8]), were more likely to become active after rest, and less likely to rest after becoming active. Furthermore, we found that even if we corrected for the average activity, the coefficient of variation, and the autocorrelation of the signals, the correlation dimension still provided additional information. Thus, it passes the first two tests of being clinically relevant. The fragmentation analysis seems to capture information that is also present in (a combination of) the average, the coefficient of variation, and the autocorrelation of the signal.

The result of our correlation dimension may be surprising as high correlation dimensions have been suggested to relate to high complexity [18] and complex dynamics have been linked to healthy subjects [46]. Do our results suggest that healthy individuals have a less complex activity pattern than DM1 patients? We believe the answer is not so simple for three reasons. First, the correlation dimension has been designed to capture the fractal dimension of a deterministic chaotic attractor. Our data is neither strictly chaotic nor deterministic. Especially measurement noise can severely limit identification of time delay embedding methods [47]. Thus, our results should not be interpreted as fractal dimensions, but instead as a number that captures the temporal organization of the signal [19,32]. Secondly, a visual exploration of the correlation integral plots of our subjects (S15 Fig) suggests that the data from healthy individuals behaves more like a chaotic attractor, and the data from the DM1 patients behaves more like red noise (see S1 Text, Sects 4 and 6.2). Thus, the differences we found might not reflect differences in 'fractal dimension' but rather differences in 'chaotic-ness' of the signals. This is actually in line with the 'healthy signals are complex' paradigm, as (depending on your exact definition of complexity), random signals are considered less complex than chaotic signals [48]. Third, even though our results suggest that the correlation dimension results are unique compared to the average activity, the coefficient of variation, and the autocorrelation, we do expect some (possibly nonlinear and synergistic) relationships between those aspects that our binary logistic regression model might not have picked up on. For example, a visual exploration (S14 Fig) of the two dimensional time delay embeddings of both groups suggests that the healthy subjects have some moments of high activity that only occur occasionally. Those episodes are not present in the time delay embeddings of DM1 patients (S1 Text, Sect 6.1). This could indicate that the correlation dimension picks up on these large excursions of activity. To further gauge the effects of these excursions, the time delay embeddings could be studied more extensively, for example with recurrence plots [15,21].

The results of the fragmentation analysis is more intuitive. Healthy individuals are more likely to become active after a period of rest than DM1 patients, and less likely to rest after active periods. This first difference is larger than the second difference, suggesting that the difference is not only linked to more periods of high activity, but also to the length of these periods of high activity, which are longer in the healthy group than in the DM1 group. DM1 patients suffer from severe fatigue and are less physically active compared to healthy controls.

In addition, patients suffer from apathy and reduced initiative compatible with the multiple periods of low activity scattered throughout the day [33,34]. This is reflected in the fragmentation results.

Since this study is an exploratory study, we used existing data from the OPTIMISTIC trial [33]. Therefore, our results and conclusions are limited by this dataset that was made available to us. One limitation is that the DM1 group was larger than the healthy group. We explored the effect of these different sizes by down-sampling from the DM1 group to obtain a sample that was equal to the healthy group with respect to age and gender (S1 Text, Sect 3). We found that the correlation dimension and kAR still had significant effects, but kRA was no longer significant. Our hypothesis is that kRA differs for both groups, but this is a small effect that requires a larger sample size. Another limitation of our study is that whereas we are interested in assessing the broad concept of 'health', we only tested our methods on healthy individuals and individuals with DM1, as two example groups where there are known differences in health. Before broader conclusions can be drawn about general complexity properties of health and disease, these indicators should be tested for multiple groups with various health conditions. Our results suggest that such an exploration is a promising avenue for future work.

Previous work suggests that activity patterns may be governed by local attractors in phase space, i.e. stable equilibria (a certain level of activity) in which the system may reside for a while [49]. A system can leave the attractor when it receives an external push or when the stability of the equilibria is altered due to either external (such as medication) or internal (such as a good nights sleep) changes. The results for our healthy group are in line with this suggestion. The correlation dimension is low, indicating regular dynamics, which could be the dynamics observed from a stable equilibrium. In contrast, our DM1 group has a higher correlation dimension and seems to behave more like red noise (S11 Fig). Therefore, it is possible that DM1 changes the underlying dynamical structure that governs an individual's activity pattern. This further supports the use of a complexity paradigm to study accelerometer data, as this paradigm can unravel (parts of) the underlying dynamical structure. This provides a possible direction to look towards when trying to explain why lifestyle interventions improve treatment of DM1 [36], or more generally, why 'multi-system' treatment approaches are important to complement treatments that focus on single systems [6,50,51].

The differences in the temporal organization of the activity patterns between the healthy and DM1 cohorts can be explained with various known differences between these groups. We controlled for the simplest ones, such as the fact that the healthy group had a higher activity on average than the DM1 group, a higher coefficient of variation and a higher autocorrelation. Another distinction between the groups that could explain our findings was differences in sleep patterns. Healthy individuals had several consecutive hours of low activity every 24 hours, whereas individuals with DM1 had multiple periods of low activity scattered throughout the day and night (linking to their high active to rest transition probability). Whereas previous studies have chosen to remove periods of sleep [9], the complicated sleep pattern of the DM1 group did not allow us to follow this approach. Also, the scattered sleep pattern is one fundamental property of the temporal organization of the activity signal that our measures might pick up on. Therefore we decided not to restrict our analysis to the daytime, but use the full day and night time series.

Our measures relate to several other techniques that are designed to extract clinically relevant information from physiological temporal data. Here, we emphasize three. Firstly, machine learning methods are currently gaining momentum to extract information about patient well-being that was previously thought to be more subjective, such as pain level experience [52]. Future work could explore combining a complex systems approach with machine

learning methods, especially in the light of making machine learning models more interpretable [53], and to make the complexity measures less restricted by their data requirements, such as even spacing between data points. Secondly, various entropy measures have shown to be indicative of fall risk in elderly people [54,55]. As visualized in Fig 2, many entropy measures such as the sample entropy are highly related to the correlation dimension, and therefore our results corroborate their finding that these complexity measures are informative to assess individual health. Lastly, 'critical fluctuations' (a temporary increase in variance) were shown to precede shifts in activity patterns [49]. This last study is particularly interesting, because the dynamical measure is used not just to distinguish between groups with known differences in health, but to actually predict the timing of a sudden gain or loss in activity. Therefore, to further assess the utility of the correlation dimension in clinical settings, future work should investigate whether changes in correlation dimensions also precede clinical diagnosis and can be used as 'early warning signals' of disease [7,56]. Investigating this potential application requires either repeated measurements at different moments in time or longitudinal data where shifts are observed [49].

## Conclusion

Our results show that the dynamical properties of accelerometer data contain clinically relevant information and that complexity measures such as correlation dimensions and fragmentation indices are well-equipped to extract this information. Furthermore, our results demonstrate that activity patterns of healthy individuals have a lower correlation dimension, a higher probability of activity after rest, and a lower probability of rest after activity than activity patterns of patients with myotonic dystrophy type I. Finally, our results support the hypothesis that activity patterns are part of a 'complex system' and that dynamical complexity measures have the potential to unravel the workings of this complex system.

## Supporting information

**S1 Text. Signals of complexity and fragmentation in accelerometer data.**
(PDF)

## Author contributions

**Conceptualization:** Els Weinans, Jerrald L. Rector, Sarah Charman, Renae J. Stefanetti, Ingrid van de Leemput, Rene Melis, Baziel van Engelen.

**Data curation:** Sarah Charman, Renae J. Stefanetti, Cecilia Jimenez-Moreno, Grainne S. Gorman.

**Formal analysis:** Els Weinans, Jerrald L. Rector.

**Investigation:** Els Weinans.

**Methodology:** Els Weinans, Jerrald L. Rector, Ingrid van de Leemput, Daniel van As, Rene Melis, Baziel van Engelen.

**Software:** Els Weinans, Jerrald L. Rector.

**Supervision:** Ingrid van de Leemput, Rene Melis, Baziel van Engelen.

**Validation:** Daniel van As.

**Visualization:** Els Weinans, Jerrald L. Rector.

**Writing – original draft:** Els Weinans, Jerrald L. Rector.

**Writing – review & editing:** Els Weinans, Jerrald L. Rector, Sarah Charman, Renae J. Stefanetti, Cecilia Jimenez-Moreno, Ingrid van de Leemput, Daniel van As, Rene Melis, Baziel van Engelen.

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
