## [Decision Letter · Decision Letter 0]

26 Jun 2025

PONE-D-24-07162Signals of Complexity and Fragmentation in accelerometer dataPLOS ONE

Dear Dr. Weinans,

Thank you for submitting your manuscript to PLOS ONE. After careful consideration, we feel that it has merit but does not fully meet PLOS ONE’s publication criteria as it currently stands. Therefore, we invite you to submit a revised version of the manuscript that addresses the points raised during the review process.

**ACADEMIC EDITOR:**

1. The authors are advised to provide more details on parameter selection in the methods section and discuss how these choices affect the results.

2. Further discussion on the finding that a lower correlation dimension may imply a more predictable pattern is recommended.

3.The potential clinical implications of these findings should be explored.

We look forward to receiving your revised manuscript.

Kind regards,

Yunhe Wang

Academic Editor

PLOS ONE

Additional Editor Comments:

1.The authors are advised to provide more details on parameter selection in the methods section and discuss how these choices affect the results.

2.Further discussion on the finding that a lower correlation dimension may imply a more predictable pattern is recommended.

3.The potential clinical implications of these findings should be explored.

Reviewers' comments:

Reviewer's Responses to Questions

**Comments to the Author**

1. Is the manuscript technically sound, and do the data support the conclusions?

Reviewer #1: No

2. Has the statistical analysis been performed appropriately and rigorously? 

Reviewer #1: Yes

3. Have the authors made all data underlying the findings in their manuscript fully available?

Reviewer #1: No

4. Is the manuscript presented in an intelligible fashion and written in standard English?

Reviewer #1: Yes

5. Review Comments to the Author

Reviewer #1: Dear Authors,

I recently had the opportunity to read your manuscript and found it intriguing, particularly from a complex systems perspective. However, I have some major concerns regarding certain technical and methodological aspects. My major concerns are outlined below.

• The sample size used for the main analysis (cross-sectional, as far as I understand from the manuscript) is very small. Additionally, one group was observed for 2 weeks, while the other group was observed for only 1 week. The sample sizes of the two groups are also notably imbalanced. The U statistic from the Mann-Whitney test is calculated based on the ranks of the combined groups. When one group is much larger, the ranks of the smaller group are spread thinly across the larger group’s ranks, which can bias the results. Finally, no power analysis was performed. While I understand that this was a convenience sample used to test the research questions, these limitations raise concerns about the reliability of the current findings.

• Higher correlations are likely descriptive of the symptoms that characterize the DM1 group—and the broader population with myotonic dystrophy type 1—rather than being indicators capable of distinguishing between groups with known health differences. Specifically, from the protocol paper (van Engelen, 2015), I understand that this population is characterized by daytime sleepiness. Therefore, one might assume that the highly irregular signals in the DM1 group are due to this condition. On a related note, in line 370 (page 15), the authors state that "Healthy individuals had several consecutive hours of low activity every 24 hours, whereas individuals with DM1 had multiple periods of low activity scattered throughout the day and night (linking to their high active-to-rest transition probability). Whereas previous studies have chosen to remove periods of sleep [9], the complicated sleep pattern of the DM1 group did not allow us to follow this approach." The fact that the complex sleep patterns prevented the authors from removing sleep periods further supports my view that the high correlation in accelerometer data might be typical of this clinical sample, rather than indicative of a generic unhealthy population. A different comparator is needed to assess the discriminative value of the correlation measures to distinguish between groups with known health differences.

6. PLOS authors have the option to publish the peer review history of their article (what does this mean?). If published, this will include your full peer review and any attached files.

Reviewer #1: No

---

## [Author Response · Author response to Decision Letter 1]

7 Feb 2025

Response is attached as `Response to Reviewers'.

---

## [Decision Letter · Decision Letter 1]

10 Apr 2025

PONE-D-24-07162R1Signals of Complexity and Fragmentation in accelerometer dataPLOS ONE

Dear Dr. Weinans,

Thank you for submitting your manuscript to PLOS ONE. After careful consideration, we feel that it has merit but does not fully meet PLOS ONE’s publication criteria as it currently stands. Therefore, we invite you to submit a revised version of the manuscript that addresses the points raised during the review process.

We look forward to receiving your revised manuscript.

Kind regards,

Sandip V George, PhD

Academic Editor

PLOS ONE

Journal Requirements:

Additional Editor Comments:

Dear Authors,

I was reassigned to this paper as academic editor. Since I am aware of the significant delays in processing this manuscript, I do not intent to send this out to additional reviewers. I do have some concerns about the manuscript, which I would like to be seen addressed in a revision. These are mostly methodological.

1. As the reviewer pointed out, the control sample lasted a week less than the DM1 sample. Correlation dimension is well known to be affected by sample length. Have the authors already checked if reducing the sample size to a week affects the differences observed? If not this would be worth investigating

2. I checked the github repo to check the above, but noticed that the code used for calculating the complexity measures were not available. Could this be added to the repo?

3. I understand that the authors interpret correlation dimension in the context of random data, but since the measure is originally developed for the analysis of signals exhibiting deterministic chaos, it would be nice to have an interpretation of what that is. In the supplementary, the authors have considered purely deterministic signals as well for their analysis. While it is generally true that regularity implies lower dimension and randomness or chaos implies higher dimensions, Correlation dimension is in essence a measure of fractality. I would recommend adding a few lines describing what exactly it is measuring in the paper, especially since a lot of the readers may be non-specialists.

4. Choice of tau=1: While in theory Taken's theorem guarantees that any value of delay time, tau, will work, for real datasets it is recommended that a delay time where the correlation (linear or otherwise) decays is chosen. Common thresholds are where the autocorrelation falls to 1/e or first minimum of the mutual information etc. It would be worthwhile to consider adding a sensitivity analysis that looks into these.

5. Choice of dimension: The authors chose dimension of 6 as far as I understand. But I don't understand Figure 6. Is this the dimension that was obtained at dimension 6? This seems unlikely, since there was no saturation observed for DM1, according to the authors. Maybe it would be worthwhile to add average D2 at every dimension for the healthy and diseased datasets in the main text/supplementary. In addition, if D2(1/2), i.e. correlation dimension at dimension 1 or 2, was chosen for the analysis, this may not be the best choice, since at this dimension the full complexity of the embedded space will not be captured by the measure.

6. Choice of threshold for transition probability analysis. Its set currently via visual inspection. Is this number close to a number that can be objectively defined? n times mean/median/standard deviation. What would be a suitable recommendation for future researchers looking to repeat this analysis for their datasets?

7. Saturation curves for example datasets: In the supplementary data, the authors provide C(R) vs R curves for the data. It would make sense to add D2(M) vs M curves as well, i.e. Correlation dimension vs dimension curves. This would make the saturation observed in the healthy controls evident.

8. Figure 11 in supplementary will be clearer if plotted using points alone. The lines obscure the structure.

Reviewers' comments:

Reviewer's Responses to Questions

**Comments to the Author**

1. If the authors have adequately addressed your comments raised in a previous round of review and you feel that this manuscript is now acceptable for publication, you may indicate that here to bypass the “Comments to the Author” section, enter your conflict of interest statement in the “Confidential to Editor” section, and submit your "Accept" recommendation.

Reviewer #1: All comments have been addressed

2. Is the manuscript technically sound, and do the data support the conclusions?

Reviewer #1: Partly

3. Has the statistical analysis been performed appropriately and rigorously? 

Reviewer #1: Yes

4. Have the authors made all data underlying the findings in their manuscript fully available?

Reviewer #1: Yes

5. Is the manuscript presented in an intelligible fashion and written in standard English?

Reviewer #1: Yes

6. Review Comments to the Author

Reviewer #1: I thank the authors for their responses to my comments. They have been clearer and more transparent about the limitations of the study. However, in my view, one of these limitations (as highlighted in my second comment during the first review round regarding the use of a different comparator) remains "unresolved." This is because the study relied on a convenience sample, and no other unhealthy comparators are available. While the authors have acknowledged this as a limitation, I remain concerned that the conclusions may be biased.

That said, given the overall rigor, novelty, and thorough documentation of the analysis, I am inclined to leave it to the editor to decide whether this limitation warrants acceptance or rejection of the article.

7. PLOS authors have the option to publish the peer review history of their article (what does this mean?). If published, this will include your full peer review and any attached files.

Reviewer #1: No

---

## [Author Response · Author response to Decision Letter 2]

23 May 2025

All reviewer and editor comments are addressed and explained in the attachment labeled as 'Response to reviewers'.

---

## [Editor Report · Decision Letter 2]

1 Jun 2025

Signals of Complexity and Fragmentation in accelerometer data

PONE-D-24-07162R2

Dear Dr. Weinans,

We’re pleased to inform you that your manuscript has been judged scientifically suitable for publication and will be formally accepted for publication once it meets all outstanding technical requirements.

Kind regards,

Sandip V George, PhD

Academic Editor

PLOS ONE

Additional Editor Comments (optional):

The authors have addressed all my queries.
---

## [Editor Report · Acceptance letter]

PONE-D-24-07162R2

PLOS ONE

Dear Dr. Weinans,

I'm pleased to inform you that your manuscript has been deemed suitable for publication in PLOS ONE. Congratulations! Your manuscript is now being handed over to our production team.

Kind regards,

on behalf of

Dr. Sandip V George

Academic Editor

PLOS ONE